# Diagnostic Challenges in Rare Causes of Arrhythmogenic Cardiomyopathy—The Role of Cardiac MRI

**DOI:** 10.3390/jpm12020187

**Published:** 2022-01-31

**Authors:** Simona Manole, Roxana Pintican, George Popa, Raluca Rancea, Alexandra Dadarlat-Pop, Romana Vulturar, Emanuel Palade

**Affiliations:** 1Department of Radiology and Medical Imaging, “Iuliu Hatieganu” University of Medicine and Pharmacy Cluj Napoca, 8, Victor Babes St., 400012 Cluj-Napoca, Romania; simona.manole@gmail.com; 2Department of Radiology, “Niculae Stancioiu” Heart Institute, 19-21, Calea Motilor St., 400001 Cluj-Napoca, Romania; 3Department of Cardiology, “Niculae Stăncioiu” Heart Institute, 400001 Cluj-Napoca, Romania; raluca_rancea@yahoo.com (R.R.); dadarlat.alexandra@yahoo.ro (A.D.-P.); 4Department of Cardiology, “Iuliu Hatieganu” University of Medicine and Pharmacy Cluj Napoca, 8, Victor Babes, St., 400012 Cluj-Napoca, Romania; 5Department of Molecular Sciences, “Iuliu Hatieganu” University of Medicine and Pharmacy, 400012 Cluj-Napoca, Romania; romanavulturar@yahoo.co.uk; 6Department of Cardiovascular and Thoracic Surgery, “Iuliu Hatieganu” University of Medicine and Pharmacy Cluj Napoca, 8, Victor Babes, St., 400012 Cluj-Napoca, Romania; paladeemanuel1@gmail.com; 7Department of Thoracic Surgery, “Leon Daniello” Pneumophtysiology Hospital Cluj-Napoca, Bogdan Petriceicu Hasdeu Street, Nr 6, 400332 Cluj-Napoca, Romania

**Keywords:** arrhythmogenic cardiomyopathy, cardio-cutaneous syndrome, ACM, ACM-LV, Padua

## Abstract

Arrhythmogenic right ventricular dysplasia (ARVD) is a rare genetic condition of the myocardium, with a significantly high risk of sudden death. Recent genetic research and improved understanding of the pathophysiology tend to change the ARVD definition towards a larger spectrum of myocardial involvement, which includes, in various proportions, both the right (RV) and left ventricle (LV), currently referred to as ACM (arrhythmogenic cardiomyopathy). Its pathological substrate is defined by the replacement of the ventricular myocardium with fibrous adipose tissue that further leads to inadequate electrical impulses and translates into varies degrees of malignant ventricular arrythmias and dyskinetic myocardium movements. Particularly, the cardio-cutaneous syndromes of Carvajal/Naxos represent rare causes of ACM that might be suspected from early childhood. The diagnostic is sometimes challenging, even with well-established rTFC or Padua criteria, especially for pediatric patients or ACM with LV involvement. Cardiac MRI gain more and more importance in ACM diagnostic especially in non-classical forms. Furthermore, MRI is useful in highlighting myocardial fibrosis, fatty replacement or wall movement with high accuracy, thus guiding not only the depiction, but also the patient’s stratification and management.

## 1. Introduction

Arrhythmogenic right ventricular dysplasia (ARVD) is a genetic condition of the myocardium, clinically characterized through syncope, malignant ventricular arrhythmias and sudden death. The pathological substrate of the disease is represented by the replacement of the ventricular myocardium with fibrous adipose tissue. ARVD is a rare condition, with a prevalence of 1:2000–1:5000 [1], more often encountered in young patients and athletes. Due to recent research and an improved understanding of the genetic substrate [1,2,3,4,5] the definition refers to a larger spectrum of myocardial involvement, which includes, in various proportions, both the right ventricle (RV) and the left ventricle (LV). The genes involved in the development of this condition are responsible, in the majority of cases, for coding desmosomes; desmosomes are complex protein structures that ensure the mechanical and electrical intercellular continuity, being located mainly in the myocytes and epidermis. Corrado et al. (2020) defined the Padua Criteria, classifying arrhythmogenic cardiomyopathy as predominant RV-sided (ACM-RV), predominant LV-sided (ACM-LV) and biventricular ACM [6].

Particularly, the cardio-cutaneous syndromes of Carvajal/Naxos represent rare causes of ACM that might be suspected from early childhood.

The diagnostic is sometimes challenging, even with well-established rTFC or Padua criteria, especially for pediatric patients or ACM with LV involvement. Therefore, the Padua system was developed to assess the involvement of LV based on new criteria, including MRI features. Cardiac MRI (CMRI) gain more and more importance in ACM dg especially in non-classical forms. Furthermore, MRI is useful in highlighting myocardial fibrosis, fatty replacement or wall movement with high accuracy, thus guiding not only the depiction, but also the patient’s stratification and management.

## 2. Genetics

According to the database of the anatomical variants of the ARVC (The ARVC Genetic Variants Database), in the last two decades more than 1400 variants of the 12 genes involved in ACM have been discovered, as follows: plakophilin (PKP2), desmoplakin (DSP), desmochollin (DSC2), desmoglein (DSG2), plakoglobin (JUP), transforming growth factor (TGFG3), transmembrane protein 43 (TMEM43), A/C lamina (LMNA), desmin (DES), titin (TTN), phospholamban (PLN), catenin alpha-3 (CTNNA3). Out of these variants, approximately 411 mutations are considered pathogenetic for ARVC spectrum [3].

Mutation screening in desmosome genes is performed as a gold standard to identify point mutation in ACM. Up to 50% of the patients have a proven genetic substrate, and only 50–60% of them are in desmosome genes PKP2, JUP, DSP, DSC2, DSG2 [7]. Desmosomes are complex, multiprotein structures, that ensure the mechanical and electrical intercellular continuity, being located mainly in the myocytes and epidermis. Their distribution explains the cardio-cutaneous involvement in Naxos and Carvajal syndromes. Other cases manifested ARVC due to mutations of genes that code extra-desmosomal structures such as PLN, TTN, LMNA, TGFB3, etc.

ACM is transmitted in an autosomal dominant manner in half of the cases, with a variable expressivity and penetrance depending on the age [3]. The forms transmitted in a recessive manner were first described as part of the cardio-cutaneous syndromes Naxos and Carvajal. Later on, it was observed that some recessive forms, occurring in non-syndromic cases, were caused either by homozygous individuals, compound heterozygotes (more than one mutation on the same gene) or digenic heterozygotes (mutations inherited from two desmosomal genes). This fact might explain the reduced penetrance of ACM through recessive or recessive-like transmission [3].

Additionally, it was observed that compound/digenic heterozygosity may express a more severe form of the disease, with an earlier onset, higher chances to develop sustained ventricular tachycardia/fibrillation and a five times higher risk of developing LV dysfunction and cardiac failure, compared with those who only have one mutation (monogenic) [2,8,9]. Furthermore, depending on the involved gene, ACM may manifest at a younger age (for DSG2, DSP genes) or at an older age (JUP gene). The variability of mutations in ACM influences the clinical presentation, which can be represented by a predominant RV involvement (classic phenotype, first described in 1982), predominant LV involvement (a phenotype first described in 2000 as part of the Carvajal syndrome) or a combination of the two (Carvajal/Naxos-like syndromes) [5,7] Table 1.

## 3. Pathophysiology

Classical features of ACM pathophysiology include cardiomyocyte loss and fibro-fatty replacement, inflammation and arrhythmogenesis. Even if half of individuals with genetically proven mutations concern the *desmosomes*, recent data suggests a more complex mechanism of *connexome* to be affected [10,11]. The connexoma comprises desmosome for maintaining of the mechanical adhesion, together with gap junctions and ion channel complexes for the electrical continuity between cardiomyocytes [12].

## 4. Imaging—Role of MRI

### 4.1. MRI Diagnostic Criteria

In terms of determining a precise morphology and kinetics, volume, mass and thicknesses of the cardiac cavities, MRI represents the gold standard. A revised diagnosis criteria have been established (revised Task Force Criteria 2010—rTFC 2010) which include the following: morphological and functional data (echocardiography and/or angiography and/or cardiac MRI), endomyocardial biopsy, electrocardiogram (EKG), arrhythmias, family history and genetic profile. The rTFC 2010 allowed an improved diagnosis of ACM with predominant RV or biventricular involvement, with a less sensitive diagnosis of the LV predominant form involvement. The latter was diagnosed due to the increased use of cardiac MRI in the last decade [2].

For a confident diagnosis, two major and one minor criterion have to be met or one major and two minor or four minor criteria. If one major and one minor or three minor criteria are met then there is a “borderline” ARVC diagnosis, Table 2.

The rTFC 2010 are also valid for the pediatric population, with the specification that inverted T wave in precordial derivations are often normal in children under 12 years old. The ACM diagnosis is rare under the age of 10 mainly due to the genetic penetrance that is age dependent [2]. The late postcontrast MRI sequences are not among MRI criteria, because of the existing inter-operator variability of appreciating LGE (late gadolinium enhancement) within the RV wall, which is very thin. However, recent studies [13,14,15] showed that LGE at the level of the LV may detect early stages of ACM-LV, even before the onset of parietal morpho-functional changes. The pattern of LGE in LV forms involves the infero-septal and infero-lateral regions, subepicardial and midmyocardial [1]. Interventricular septal (IVS) enhancement is present in ≥50% of predominately ACM-LV cases, unlike the forms with predominately RV involvement, in which IVS enhancement is not characteristic [7].

In order to improve the diagnosis of ARVC with predominant LV or biventricular form, Corrado et al. [6] proposed the Padua Criteria 2020. Table 3 The authors described that LGE is present more frequently in the subepicardial region compared to midmyocardial region of LV, the most affected segments being inferior and lateral ones, with or without IVS involvement. The affected areas undergo fibrotic/fibro-adipose remodeling responsible for wall akinesia or hypokinesia. Due to the fact that a significant proportion of patients do not show systolic disfunction of LV (LVEF) or wall kinetic abnormalities, LGE is decisive in the diagnosis of ACM-LV or biventricular form.

### 4.2. Differential Diagnosis—Normal Variants

The MRI differential diagnosis includes anatomical variants and pathological conditions. As a general rule, the diagnosis sensitivity is improved by correlating more diagnosis parameters, such as echocardiography, EKG, endomyocardial biopsy, cardiac electrical mapping, family history) [2]. The classical “dysplasia triangle” described for affected RV includes the focal alteration of the myocardial kinetics in the sub-tricuspid area, RV apex and right ventricle outflow tract (RVOT). However, the area of the RV situated between the moderator band insertion and the RV apex highlight a significantly variable contractility pattern in normal individuals [7]. The RV dilatation in the area of the moderator band has been described as the “butterfly” aspect of the RV apex (butterfly apex). Considering these anatomical variants and the study of Te Riele et al. in 2013 [16], when diagnosing ACM it is preferable to search for focal kinetic RV alterations in the basal segments of inferior and anterior walls.

Another anatomical variant is found in patients with pectus excavatum. Due to the rotation and displacement of the heart to the left, the sternum comes into close relationship with the right atrium and the base of the compressed RV looking like a small hypokinetic base, while the midmyocardial and apex areas have a pseudo-dilated appearance, mimicking dyskinesia. In this particular situation, the key to a correct interpretation is recognizing the pectus excavatum [7].

Tethered RV may be responsible for the third normal variant. It represents a relatively small area of systolic-diastolic dilatation of the lateral RV wall, secondary to the presence of a pericardial connective tissue membranes that tether the RV to the posterior side of the sternum [7]. The diagnosis in this case is suggested by the static aspect of the tethered area both in the systole as well as in the diastole. In case of equivocal diagnosis, CT may provide a better spatial resolution and point out the retrosternal connective tissue membrane.

### 4.3. Differential Diagnosis—Pathologic Conditions

Various pathological conditions may mimic ACM, such as RV infarction, DCM, sarcoidosis, myocarditis, idiopathic ventricular tachycardia of the RVOT, Brugada syndrome or athlete’s heart.

RV infarction implies a vascular territory and focal hypokinesia and dyskinesia are not characteristic. Additionally, LGE starts subendocardial unlike ARVC which is subepicardial/midmyocardial.

Patients with ACM-LV involvement require a differentiation from DCM (especially the non-ischemic type). Most patients with DCM do not show LGE. In those who do present LGE, the uptake is midmyocardial or subepicardial in the non-ischemic type, similar to ACM-LV. Furthermore, DCM cases do not present involvement of the RV, while ACM-LV is associated with ventricular malignant arrhythmias more often and at a much earlier stage than DCM [2,7].

Sarcoidosis affecting the heart may mimic the aspect of ACM. The personal history, and the presence of mediastinal and/or lung hilum adenopathy with lung infiltrates in different stages of the disease are criteria for differentiation. Patients with isolated cardiac sarcoidosis involvement are rare [17], and the diagnosis based exclusively on the MRI and CT aspect is not recommended. Furthermore, PET or Gallium scintigraphy are able to highlight the activity of sarcoid granulomas being useful in diagnostic as well as conducting the treatment. Classically, sarcoidosis affects the base of the IVS, where due to the non-caseous myocardial necrosis, mural thinning and focal dyskinesias develops. It is acknowledged that sarcoidosis may deteriorate the atrioventricular node, thus explaining the high grade atrioventricular block. The new guidelines proposed by the Japanese Circulation Society (JCS) in 2017 consider the morpho-functional involvement of the IVS base, the contractile dysfunction of the LV (LVEF ≤ 50%) and the presence of LGE as being major criteria in diagnosis of sarcoidosis. In equivocal situations, additional examinations are recommended (e.g., endomiocardial biopsy and cardiac electrical mapping).

Myocarditis has a highly variable clinical presentation but often is similar to that of heart failure. Occasionally it may show clinical features similar to ARVC. The differential diagnosis based on the LGE is inconclusive as they are very much alike. The STIR sequence may reveal focal/diffuse intramyocardial edema in acute/subacute myocarditis, but there are cases of ACM that also show intramyocardial edema, the aspects being difficult to differentiate [18]. That is why it is mandatory to correlate these aspects with the clinical, biological, nuclear medicine and cardiac electrical mapping findings or with endomyocardial biopsy.

Idiopathic ventricular tachycardia of RVOT is a typically benign condition [2]. An accurate differentiation among this entity and ACM is made when the latter displays a completely expressed phenotype. The ACM’s morpho-functional alterations described in rTFC 2010 are usually not present in the case of idiopathic ventricular tachycardia of RVOT [19]. However, the differentiation between early forms of ACM and idiopathic ventricular tachycardia is difficult, requiring additional evaluations.

Brugada syndrome is an arrhythmogenic cardiac manifestation first described in 1992, which is responsible for ventricular tachycardia and sudden death. The Shanghai Criteria has been proposed for the diagnosis of Brugada syndrome, usually in the presence of a type 1 ECG [20]. There is overlap with ACM considering the ventricular fibrotic changes, dilatation of RVOT, RV focal dyskinesia, and even the presence of common gene mutations (sodium channel 5A—SCN5A). However, patients with Brugada syndrome do not exhibit global RV systolic dysfunction or global dilatation [20], and one study showed that only 8% presented LGE and none met rTFC 2010 [21].

A particular case that requires a differential diagnosis is the heart of the athletes. Intense training leads to physiologic cardiac adaptation such as moderate dilatation of the RV and LV (with RV/LV index kept under 0.9), EKG anomalies and arrhythmias, but without regional or global systolic dysfunction. After training, ventricular volume usually decreases. Furthermore, an athlete’s heart does not present fibrotic changes on MRI. The imaging aspects that favor the ARVC diagnosis are the global systolic dysfunction and/or focal dyskinesia (systolic +/− diastolic bulging) and LGE (presence of fibrosis) [2,22].

## 5. Cases Presentation

**First patient:** H.C-D, 21 years old, presents into the emergency room for a syncope that occurred while standing, palpitations and moderate effort dyspnea. The family history revealed that the patient had a sister who suffered sudden death at the age of 21 (minor criteria for ACM). The clinical examination showed woolly hair and hyperkeratotic lesions. Figure 1.

The EKG showed flattened T waves in the limb leads and negative in V5 and V6, ventricular extrasystoles with left bundle branch block (LBBB) and right bundle branch block (RBBB) (Figure 2).

Echocardiography (ECG) described RV dilatation (57/49 mm), hyperechoic interventricular septum (IVS) and a severely reduced EF (LVEF = 29%). Microaneurysms were present in the IVS. The lateral wall of the RV revealed noncompaction of the myocardium. RV presented hypokinetic areas at the apex, the inferior wall and the RV outflow tract (RVOT—37 mm PLAX view; 47 mm PSAX view) and a small dyskinetic area at the apex (major ARVC criteria). The systolic function of the RV was very reduced (RVEF = 29%). The delayed ventricular potentials monitored through SAECG detected 3 positive parameters (filtered QRS duration = 137 ms (>114 ms); root-mean-square of the last 40 ms of QRS = 4μV (<20 μV), low-amplitude signal duration of QRS terminal = 53 ms (≥38 ms), which represents a minor criterion for ARVC, Figure 3. A 24 h Holter revealed, while on a sinus rhythm, 5922 VES (ventricular extrasystoles), polymorphic, isolated, coupled, triple and 20 episodes of unsustained, polymorphic ventricular tachycardia (>500 VES/24 h is a minor criterion for ARVC).

The CMRI highlighted the following: biventricular dilatation (LV = 52/58 mm; RV = 53 mm short axis base), global hypokinesia of the LV and RV, with small akinetic area in the IVS (Figure 4D), focal dyskinetic areas in the RVOT (Figure 4A), anterior wall of the LV (Figure 4B) and RV apex (Figure 4C). LGE confirmed the extended biventricular fibrosis of the myocardium, which showed a midmyocardial, concentric distribution in the LV, starting from the basal up to the apical segments (Figure 4E,F). Focal contrast enhancement alternating with areas of normal myocardium were noticed in the walls of the RV. The ventricular function is severely altered (LVEF = 22.6%; RVEF = 27.2%, EDV/S = 123 mL/m^2^)—major MRI criteria for the diagnosis of ARVC.

As a result of the clinical and laboratory evaluation, the diagnosis of ARVC, as part of a cardio-cutaneous syndrome, was established with certainty (two major and three minor criteria rTFC 2010). Genetic testing revealed the presence of three heterozygotic mutations of the DSP gene (c.313C>T, c.88G>A, c.273+5G>A)—the patient is compound heterozygote. The parents were carriers with one and two mutations of the DSP gene, respectively, and they were clinically and paraclinically healthy; it must be mentioned that both the mother, as well as the deceased sister, had woolly hair. The final diagnosis was that of ARVC in the context of a Carvajal/Naxos-like syndrome.

**The second patient,** F.D.S., 10 years old, is brought to the outpatient for effort dyspnea. Woolly hair and palmoplantar keratoderma were observed with the latter affirmative present since the patient was one year old (Figure 5).

He has a sinus rhythm, 86 bpm and a grade I/II systolic murmur; echocardiography reveals a dilatation of the left cavities (LV = 54.8/48.4 mm), diffuse hypokinesia of the LV and very low EF (LVEF = 25%); low amplitude mitral and pulmonary regurgitation, patent foramen ovale. The EKG showed delta wave and a short PS interval compatible with Wolf–Parkinson–White pre-excitation syndrome. A 4 h Holter EKG was performed (insufficient), detecting 244 ventricular extrasystoles.

A cardiac MRI showed the following: dilated LV (51/60 mm), non-dilated RV (41 mm short axis) (Figure 6A), global hypokinesia of the LV and RV, more severe at the level of IVS, without dyskinetic areas. The pattern of the LGE revealed inhomogeneous midmyocardial fibrosis of all the walls of the LV in the basal, mid and apical segments, with scattered area of normal parenchyma (Figure 6B,C). The ventricular function is severely altered (LVEF = 23%; RVEF = 25%; EDV/S = 72 mL/m^2^).

The rTFC 2010 criteria were not met in this case. However, the Padua Criteria for ACM-LV “borderline” were met (one major and one minor). Considering the association between the cardiac involvement and the cutaneous changes in such a young patient, a cardio-cutaneous syndrome could not be excluded. Genetic testing found two heterozygotic mutations of the DSP gene (c.7566_7567delinsCp.R2522Sfs*39, c.7756C > T, p.R2586*)—therefore, the patient is a compound heterozygote. The parents did not manifest the symptoms of the patient, each being a carrier of a heterozygotic mutation of the gene. The final presumable diagnosis is ACM in the context of the Carvajal cardio-cutaneous syndrome.

**The third patient**, L.S.-P., 27 years old, presented to the outpatient clinic for daily, low intensity arrhythmias for the last 10 years, with normal tolerance to effort. A cardiac examination showed the following: 100 bpm, normal rhythm cardiac murmurs, no pathological cardiac murmurs associated. An EKG revealed the following pathological alterations: sinus arrhythmia, corrected QT interval = 451 ms (slightly elevated, NV ≤ 440 ms in men), quadrigeminal VES. Echocardiography showed hypokinesia of the infero-lateral and antero-lateral walls of the LV in the mid-apical segments, base and mid of the IVS and hypokinesia of the RV inferior wall and RVOT (34 mm in PLAX view, 32 mm in PSAX view). The 24 h Holter examination detected 10,002 bigeminal and trigeminal VES (minor criteria for ACM).

Cardiac MRI detected enlarged right cavities and dilatation of the LV (RA = 55/49 mm, RV 58/46 mm, LV = 63/55 mm). There is also concentric LV hypokinesia beginning from the mid segments towards the distal apex and mild global RV hypokinesia associated with a few small, focal dyskinetic areas (systolic bulging) within the RVOT (Figure 7A) and the free wall (Figure 7B), just above the insertion of the moderator band. Functionally, the RV presents a moderate contraction dysfunction (RVEF = 34.6%). The last two aspects represent one major criteria for the MRI diagnosis of ARVC. LVEF is mildly decreased (55.6%, NV > 56% for age). There is discrete LGE, suggestive for myocardial fibrosis, appearing as focal area within the inferior wall (Figure 7C), the lateral wall (more obvious, Figure 7C,D), the RV apex and in lower degree the RVOT. Medioapically speaking, there is a hypertrabeculated pattern of both ventricles, and in the LV the aspect is between the highest value for normality versus noncompaction. Genetic testing revealed mutation of TMEM-43 gene. According to the rTFC 2010 (three major and one minor) and according to Padua Criteria 2020 (four major and one minor), the criteria were met and this is a definitive diagnosis for ACM-RV, without the association of cutaneous syndrome.

## 6. Discussion

Patients 1 and 2 are the first two cases of ACM as part of the Carvajal syndrome transmitted in a recessive manner from Romania. Patient 3 manifests a classic, non-syndromic form of ACM. The patients were chosen for this presentation to underline the fact that ACM is not an RV exclusively disease as it was considered in the past, but a specter of conditions with various degrees of ventricular myocardial involvement as follows: predominately RV involvement (patient 3), predominately LV involvement (patients 2), biventricular involvement (patient 1) as part of a syndromic inheritance or not. Furthermore, the role of CMR is vital, especially in patient 2, where only Padua criteria were met for a final diagnosis.

The cardiac involvement in the context of Carvajal syndrome in patients 1 and 2 consists of an ACM phenotype with a pattern of midmyocardial fibrotic changes (similar to idiopathic DCM) [23], the aspect being in agreement with literature data [24].

Patient 1 has two out of three DSP gene mutations (c.88G>A and c.273+5G>A) which are included in the database of the genetic variants of ARVC (last checked on 20.12.2021). According to Bauce et al. [25], these mutations have been found present in patients with known ACM, but not in the healthy population.

The second patient’s DSP gene mutations are not included in the database of the genetic variants of ACM. The case has been reported initially by Pigors et al. [26]; at that time (age 5) the patient was reported as having no cardiac anomalies, but with focal keratoderma on palms and soles. Interestingly, the patient was also reported as having hypotrichosis; at the age of 10, he developed cardiac abnormalities and he had woolly hair.

We included patient 3’s RV apex contractility pattern under dyskinetic changes. Although Te Riele et al. [7] pointed out the possibility of it being a “butterfly apex”, normal anatomical variants of the RV contractility pattern do not include dyskinetic changes seen elsewhere in the RV, and rTFC 2010 are not met, as seen in this case.

Patients with Carvajal syndrome present with symptoms from a young age, as the cardiac involvement can be severe before 20; the mean age of death is 14, most patients dying as a result of severe cardiac failure or sudden death due to malignant ventricular arrhythmias. Our first two patients demonstrated severe left cardiac involvement at the ages of 21 and 10.

The patients were diagnosed according to rTFC 2010. As previously mentioned, these criteria are insufficient for diagnosis because of the following two major reasons: they do not address the patients with LV involvement and there is no diagnosis variant for the pediatric population. It is a well-known fact that children under 12 years old can physiologically manifest inverted T waves in the precordial leads, which is why the major criteria for repolarization cannot be taken into account. The multitude of cases with predominant left side involvement is due to the more frequent use of cardiac MRI in the last 10 years. Where rTFC 2010 were not met, we took in consideration the Padua Criteria 2020 proposed by Corrado et al., with new criteria assessing the LV involvement, and observed an improved diagnostic, highlighting the central role CMR has in this rare cause of ACM.

We did not mention the fatty changes within the myocardium due to the fact that up until now, no consensus has been established regarding the imaging differential diagnosis between physiologic myocardial lipogenesis and the one developing in the context of ACM. Its appearance at a young age does not represent a solid differentiation criterion. Cannavale et al. [27] proposed the following elements for the differential diagnosis: fatty infiltration of the free wall of the RV in association with parietal thinning (<2 mm) in favor of pathological substrate of lipogenesis. Our cases had some limitation of note, regarding a relatively low spatial resolution which prevents an accurate appreciation of the lipomatous infiltration of the RV and its parietal thickness.

The differential diagnosis with myocarditis is difficult in children and teenagers. ACM may manifest in these age groups in a so-called “warm phase”, before the onset of fibrosis, when the progression of the disease becomes accelerated, with biventricular morphological and functional alterations. Intramyocardial edema detected on the STIR sequence may overlap with the aspect of myocarditis [28,29]. A possible element for the differential diagnosis was proposed by Martins et al. [28], who suggested that ACM must be considered in cases of multiple “myocarditis-like” episodes occurring in the same patient. Patient number 2 did not manifest certain myocardial edema on the cardiac MRI examination.

An interesting aspect was the association of ACM with the CMR findings of ventricular noncompaction in the situation of patient number 1 and number 3. Arbustini et al. [30] illustrated the specter of ventricular noncompaction including the possible association with ACM among other things. Additionally, genes that are responsible for ACM were also found in patients with ventricular noncompaction, as follows: DSP, PKP2 genes and the ryanodine 2 receptor (RYR2). Indeed, patient number 1 presents the Carvajal/Naxos-like syndrome in association with ventricular noncompaction and the presence of the DSP mutation; there are about five published cases that manifest this association [31]. Patient number 2 did not manifest this association, but it must be taken into account that it was a pediatric case. It is not known whether ACM and noncompaction appear independently, being just genetically conditioned or whether noncompaction develops during the evolution of ACM.

Even though noncompaction is often described as a condition with intrauterine onset and a predominantly non-desmosomal genetic substrate, in particular cases when ACM has a desmosomal substrate, the possibility of common pathogenesis mechanism can be raised; under the intracavitary mechanical pressure and the alteration of the intercellular signaling [1], the myocytes disassemble, undergo lipogenesis and afterwards fibrosis. This disassembling may also happen in the shape of trabecular bundles, resulting in the hyper-trabeculated aspect. This assumption deserves to be studied in the future by performing a CMR follow-up of the young patients with ACM that show no hyper-trabeculation, in order to detect its possible development over time, since it is known that there is an additional risk of cardiac sudden death in young individuals associating DCM and noncompaction [32]. It would also be useful to perform genetic testing for desmosomal mutations at a young age, in individuals with ventricular noncompaction diagnosed peripartum, in order to discover cases with possible ACM with a genetic substrate, thus being able to treat the condition in a preclinical stage.

## 7. Conclusions

ACM represents a myocardial condition that involves both the RV as well as the LV in various degrees. The patients are usually young, and the prognosis is poor in undiagnosed cases, with death occurring due to malignant ventricular arrhythmias. The CMR examination brings a significant contribution to the diagnosis of ACM, especially in cases with predominately LV involvement, where the delayed post contrast PSIR sequence is essential. Furthermore, CMR may help distinguishing ACM from normal or pathologic variants.

## Figures and Tables

**Figure 1 jpm-12-00187-f001:**
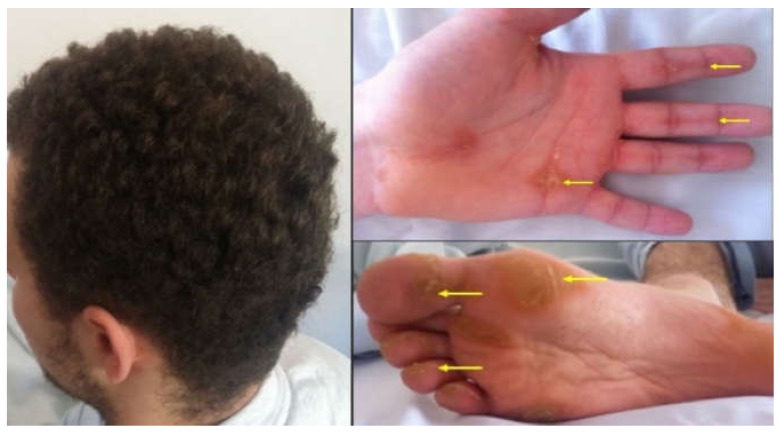
The patient has woolly hair, characterized by the fact that the hair is curly and dull. Striated fibrous palmar hyperkeratosis—hyperkeratotic lesions are located on the fingers and flexion areas (arrows). Focal plantar hyperkeratosis—hyperkeratotic lesions are located in the pressure zones (arrows).

**Figure 2 jpm-12-00187-f002:**
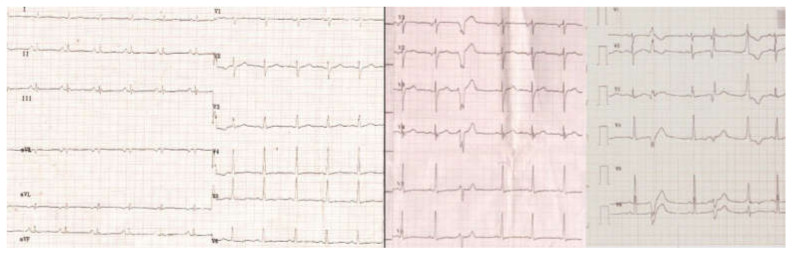
The ECG shows hypovoltage, nonspecific intraventricular conduction disturbances, negative T-waves in V5-V6 and polymorphic ventricular extrasystoles.

**Figure 3 jpm-12-00187-f003:**
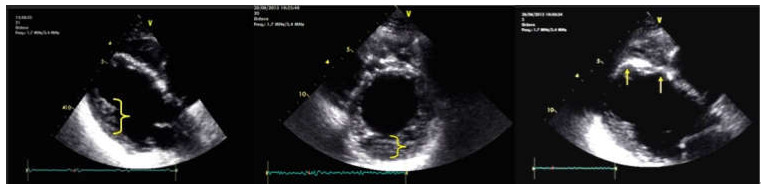
Cardiac ultrasound. Left image: long-axis parasternal section shows dilated LV, with hyperechoic thin SIV, and LV noncompaction areas at the posterolateral wall (brace); middle image: short axis parasternal section displays noncompact LV area (brace) > 2 × compact LV area; right image: parasternal section long axis shows aneurysms located at the SIV level (arrows).

**Figure 4 jpm-12-00187-f004:**
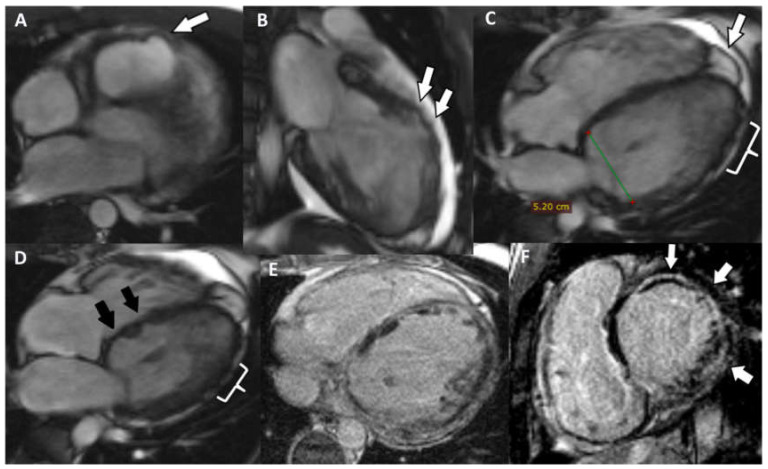
(**A**) Axial Cine-GRE sequence: focal dyskinesia of the RVOT; (**B**) Cine-GRE two-chamber LV sequence: focal dyskinesia of the anterior wall of the LV; (**C**,**D**) four-chamber Cine-GRE sequence: focal dyskinesia of the RV apex (arrow), dyskinesia of the lateral wall of the LV (square brackets), akinetic IVS areas (black arrows); (**E**) PSIR sequence—delayed postcontrast axial view: concentric midmyocardial LGE within the LV and diffuse, inhomogeneous LGE of the lateral wall of RV; (**F**) PSIR sequence—delayed postcontrast short axis view: midmyocardial LGE of the LV base.

**Figure 5 jpm-12-00187-f005:**
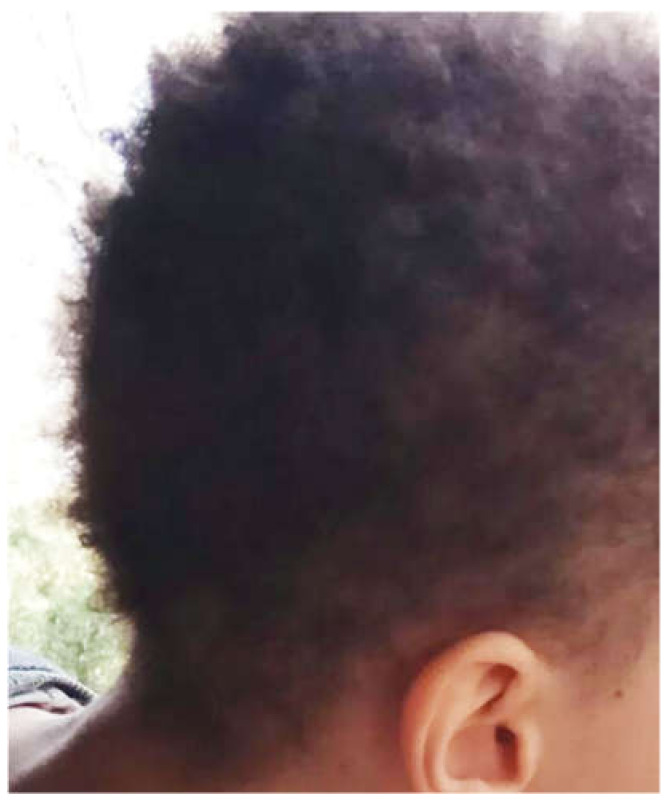
Woolly hair in a 10-y old patient.

**Figure 6 jpm-12-00187-f006:**
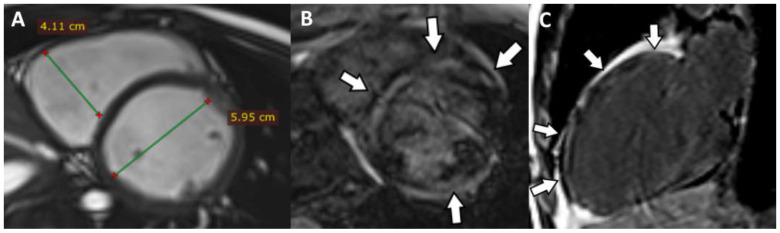
(**A**) Short axis Cine-GRE sequence: basal dilatation of the LV, normal size RV; (**B**) short axis delayed postcontrast PSIR sequence: concentric midmyocardial LGE of LV; (**C**) LV two chambers delayed postcontrast PSIR sequence: non-uniform midmyocardial LGE of the anterior wall of the LV.

**Figure 7 jpm-12-00187-f007:**
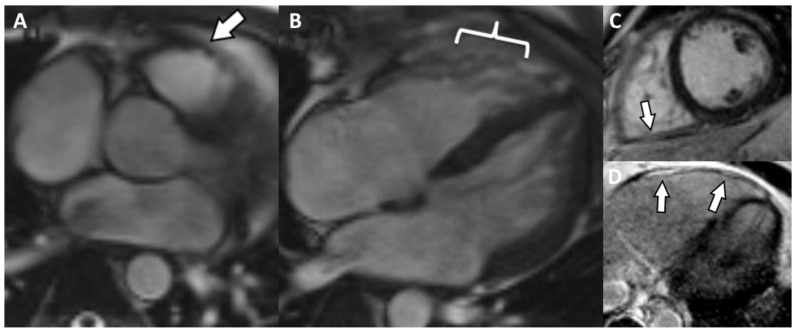
(**A**) Axial Cine-GRE sequence: focal dyskinesia of the RVOT; (**B**) four-chamber Cine-GRE sequence, systole: focal dyskinesias of the lateral wall of the RV, above the insertion of the moderating band; (**C**) PSIR delayed postcontrast sequence short-axis: diffuse LGE of the lateral wall of the RV, focal LGE of the inferior wall (arrow); (**D**) Axial delayed postcontrast PSIR sequence: LGE of the lateral wall of the RV.

**Table 1 jpm-12-00187-t001:** Affected gene and corresponding ventricular involvement on ACM.

Predominant RV Involvement	Biventricular Involvement	Predominant LV Involvement
JUP	TMEM43	DSP
PKP2		DES
DSG2		PLN

**Table 2 jpm-12-00187-t002:** The revised Task Force Criteria 2010 for ARVC.

MRI Criteria	Description
Major	Focal akinesia/dyskinesia of the RV/asynchronous ventricular contraction and one of the following: RV dilatation (EDV index ≥ 110 mL/m^2^ men, ≥100 mL/m^2^ women) RVEF ≤ 40%
Minor	Focal akinesia/dyskinesia of the RV/asynchronous ventricular contraction and one of the following: RV dilatation (EDV index ≥ 100 but <110 mL/m^2^ men, ≥90 but <100 mL/m^2^ women) RVEF ≤ 45% but >40%

**Table 3 jpm-12-00187-t003:** Padua 2020 criteria for LV involvement.

MRI Criteria	Description
Major	LGE ≥ 1 segment (in two orthogonal views) of the free wall (subepicardial or midmyocardial), IVS or both
Minor	Depression of LVEF with or without dilatation of LV, according with data from age, sex, and body surface are nomograms Regional akinesia or hypokinesia of the free wall, IVS or both

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
