# Peer review of "Diagnostic Challenges in Rare Causes of Arrhythmogenic Cardiomyopathy—The Role of Cardiac MRI"

_jpm, 2022, doi:10.3390/jpm12020187_

Round 1

Reviewer 1 Report

In the present manuscript, Manole et al. present a review on various difficulties in the diagnosis of rare arrhythmogenic cardiomyopathy and present cardiac MRI technique as a potentially useful solution for diagnosis by providing case studies.

This is an important review manuscript contributing toward better understanding of the diagnostic challenges and potential solution for rare cases of arrhythmogenic cardiomyopathy. 

Minor comments:

  • As this is a review paper, a graphical representation with a flowchart or figure of current diagnostic techniques available in a more clearly arranged way would be very helpful to the readers.

  • CMR (Cardiac MRI) full form should be provided at first use on line 238.

  • The title reads “…rare causes of arrhythmogenic cardiomyopathy…” should it be “…rare cases of arrhythmogenic cardiomyopathy…”?

Author Response

Thank you very much for your time and comments.

Minor comments:

  • As this is a review paper, a graphical representation with a flowchart or figure of current diagnostic techniques available in a more clearly arranged way would be very helpful to the readers
  • Thank you for the suggestion, we also believe that a flowchart with diagnostic techniques is helpful; we provided one version. It is not clear for us, if the reviewers could evaluate it, because it is not included in the main manuscript. Looking forward to your response.

  • CMR (Cardiac MRI) full form should be provided at first use on line 238.  Thank you for this observation. We used cardiac MRI for the first time within the text, at line 56; we added the abbreviated form. Please let us know if anything else should be added on line 238 which is describing EKG and echocardiography.

  • The title reads “…rare causes of arrhythmogenic cardiomyopathy…” should it be “…rare cases of arrhythmogenic cardiomyopathy…”? 
  •  Indeed, we present rare cases of arrhythmogenic cardiomyopathy, but we believe they also represent rare diseases that causes or lead to cardiomyopathy, reason why we would like to keep the ‘’causes’’ instead of cases. Please let us know if that suits you.

Reviewer 2 Report

The paper is well organized and explained the diagnostic procedure of ARVD and ACM clearly.I think it is a review helping people understand the mechanism of the disease and how to diagnose it .

For Table II and III, I think the authors can consummate the tables by referring  to table I, in the form of lines.

Figures 4~17 can also be rearranged for better understanding.

Author Response

The paper is well organized and explained the diagnostic procedure of ARVD and ACM clearly.I think it is a review helping people understand the mechanism of the disease and how to diagnose it . Thank you very much for the appreciation.

For Table II and III, I think the authors can consummate the tables by referring to table I, in the form of lines

We updated the Table II and III.

Figures 4~17 can also be rearranged for better understanding --

The figures appear in the order in which they are referred within the text. Perhaps we can merge them in order to have fewer images with figure parts (eg. Fig 4ab)? Please let us know your thoughts on that.